# Development of protective equipment for endoscopic treatment and surgery in otorhinolaryngology

**Yoshinori Matsuwaki** [1]*, **Daiki Ariyoshi**[2], **Risa Iwamoto**[2], **Tetsuya Morishima**[2], **Keita Kimura**[2], **Daisuke Kikuchi**[3]

1 Matsuwaki Clinic Shinagawa, Tokyo, Japan, 2 Olympus Medical Systems Corporation, Tokyo, Japan, 3 Department of Gastroenterology, Toranomon Hospital, Tokyo, Japan

* yoshi@matsuwaki.com

## Abstract

### Objective

The coronavirus disease pandemic has raised concerns regarding the transmission of infections to healthcare workers. We developed a new protective device to reduce the risk of aerosol diffusion and droplet infection among healthcare workers. Here, we report the results of a theoretical evaluation of the efficacy of this device.

### Methods

We used suction-capable masks with and without rubber slits, sleeves for the insertion section of endoscopes and treatment tools, and a cover for the control section of the endoscope. To simulate droplet spread from patients, we created a droplet simulation model and an aerosol simulation model. The results with and without the devices attached and with and without the suction were compared.

### Results

The droplet simulation model showed a 95% reduction in droplets with masks with rubber slits; furthermore, a reduction of 100% was observed when the insertion sleeve was used. Evaluation of aerosol simulation when suction was applied revealed an aerosol reduction of 98% and >99% with the use of the mask without rubber slits and with the combined use of the mask and insertion sleeve, respectively. The elimination of droplet emission upon instrument removal confirmed that the instrument sleeve prevented the diffusion of droplets. The elimination of droplets upon repeated pressing of the suction button confirmed that the cover prevented the diffusion of droplets.

**Data Availability Statement:** All relevant data are within the manuscript.

**Funding:** The authors received no specific funding for this work.

**Competing interests:** This research and development endeavor was conducted in collaboration with Olympus Medical Systems Corporation and a gastrointestinal endoscopist (Dr. Daisuke Kikuchi). Matsuwaki Clinic Shinagawa and Olympus Medical Systems Corporation have concluded an advisory agreement regarding this development. This does not alter our adherence to PLOS ONE policies on sharing data and materials.

## Conclusion

We developed a device for infection control, in collaboration with a gastrointestinal endoscopist and Olympus Medical Systems Corporation, that was effective in reducing droplet and aerosol diffusion in this initial theoretical assessment.

## Introduction

The global outbreak of the new coronavirus disease 2019 (COVID-19) has raised concerns regarding the number of patient deaths and transmission of infection to healthcare workers. The severe acute respiratory syndrome coronavirus 2 (SARS-CoV-2), the virus responsible for COVID-19, has been detected at the highest concentrations in sputum and upper respiratory tract secretions [1, 2]. In addition, the virus can be transmitted by and remains infectious even in small droplets [3]. Virus-containing droplets vary in size, with larger droplets (>60 μm in diameter) tending to fall on surfaces closer to the patient (<2 m) and smaller droplets (10–60 μm in diameter) tending to fall on more distant surfaces [4]. It is known that aerosolized droplets with a diameter of <5 μm can become airborne [5, 6].

Many studies have shown that the virus can be detected in the air after infected patients cough, sneeze, or exhale in indoor environments [7, 8]; therefore, the risk of infection to healthcare workers is high. In particular, aerosol-generating treatment procedures are thought to increase the risk of infection to healthcare workers [1, 2]. It is known that the possibility of inhaling viruses in respiratory droplets is very high at a distance of a few meters, as is the case in examination and operating rooms [3]. In fact, there is much evidence that aerosols and droplets are generated during outpatient care, surgical care, and procedures in otorhinolaryngology [2, 9–12].

The risk of infection by such aerosols and droplets is not limited to otorhinolaryngology but is also a concern in other fields, such as gastrointestinal endoscopy [13–16], and personal protective equipment (PPE) for infection prevention is becoming increasingly important for healthcare workers.

We have developed a new infection control device for use in outpatient otorhinolaryngology and surgery clinics in collaboration with a gastrointestinal endoscopist and Olympus Medical Systems Corporation. In this report, we describe the use of this infection control device and its effectiveness in reducing the spread of droplets and aerosols.

## Materials and methods

### Device

We developed the following items: a mask to be worn by the patient, an insertion sleeve to be placed over the insertion section of the endoscope, an instrument sleeve to be placed over the instrument, and a control section cover to be placed over the control section of the endoscope (Fig 1). The mask has four ports on its sides that allow air inflow and suction of aerosols and droplets generated by the patient. The ports can be equipped with a cap that can change the direction of air inflow, and a connector that can be connected to a suction tube. An endoscope insertion port is located in front of the mask, and this can be connected to the insertion sleeve with a joint. We developed two types of masks: one for outpatient use that contains a rubber slit at the endoscope insertion port to prevent splashes caused by coughing (Fig 1A), and one for surgical use with an open endoscope insertion port so that operability is not affected even

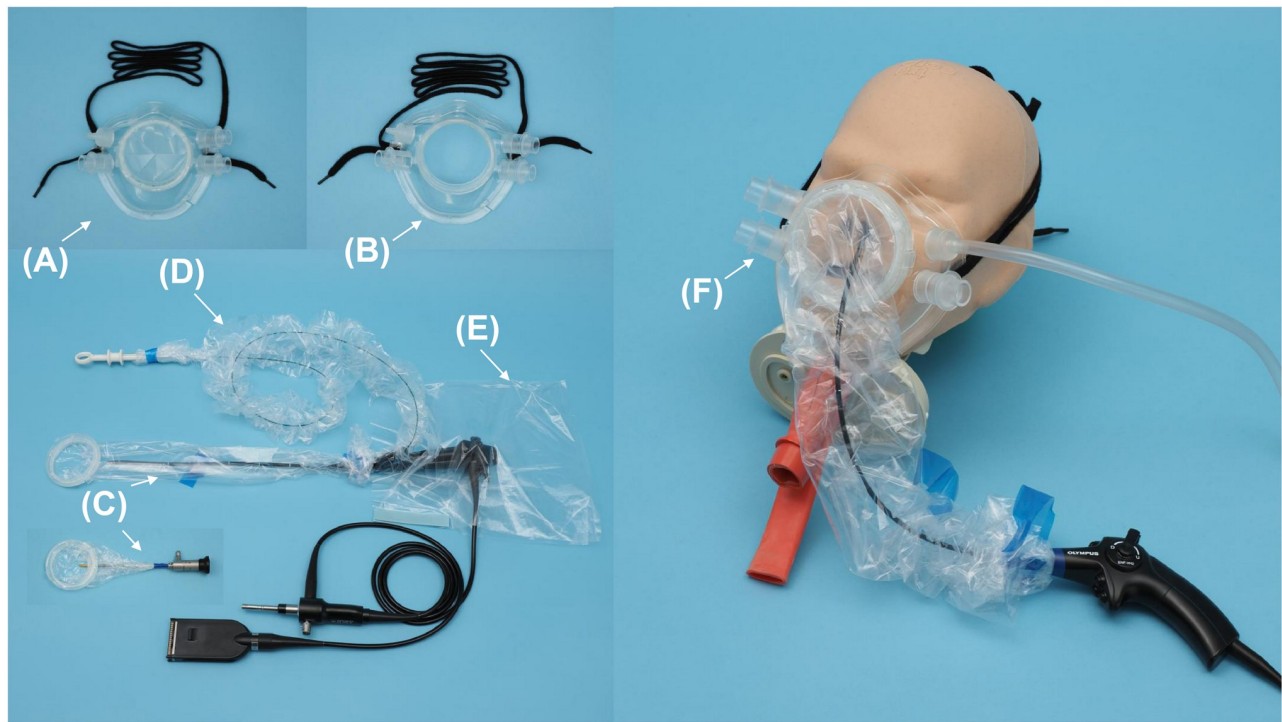

**Fig 1. Images of the prototype product.** (A) A mask with a rubber slit (for outpatient use). (B) A mask without a rubber slit (for surgery). (C) Insertion sleeve (attached to a flexible endoscope [Olympus ENF-VT3; Olympus Medical Systems Corporation, Tokyo, Japan] and a rigid endoscope [WA4KA400; Olympus Medical Systems Corporation, Tokyo, Japan]). (D) Instrument sleeve. (E) Control section cover. (F) An example of how the mask and insertion sleeve are attached to the head model when using a flexible endoscope for observation (OLYMPUS ENF-VH2; Olympus Medical Systems Corporation, Tokyo, Japan). A suction tube can be connected to the suction port of the mask. The endoscope insertion port of the mask and the insertion sleeve can be connected by a joint.

when multiple instruments, such as a rigid endoscope and operative tools, are inserted (Fig 1B). These masks can also be worn over the nasal cannula and intubation tube. The insertion sleeve can cover the insertion section of the endoscope and can be attached to both rigid and flexible endoscopes. The insertion sleeve consists of a part that covers the insertion section of the endoscope and a joint that can be connected to the mask and it is used by fixing it at the root after inserting the endoscope insertion section (Fig 1C). The instrument sleeve has a part that covers the instrument and can be fixed near the handle of the instrument after insertion. The instrument can be inserted into the instrument channel with the sleeve attached, and one end of the instrument sleeve is fixed with a hook and loop fastener to the endoscope (Fig 1D). The control section cover is used to cover the control section of the endoscope. It has a hole for the hand of the endoscope operator, the universal cord of the endoscope, and the insertion section of the endoscope, and is used by fixing it with a hook and loop fastener after covering the control section (Fig 1E). An example of the mask and insertion sleeve on the head model is shown in Fig 1F.

## Device use

The procedure for use of the device during outpatient care and surgery, respectively, is as follows. First, during outpatient examinations, the insertion sleeve is affixed to the endoscope and fixed at the root. The patient puts on the mask. A suction tube is attached to the port of the mask. The joint of the insertion sleeve is connected to the endoscope insertion port of the

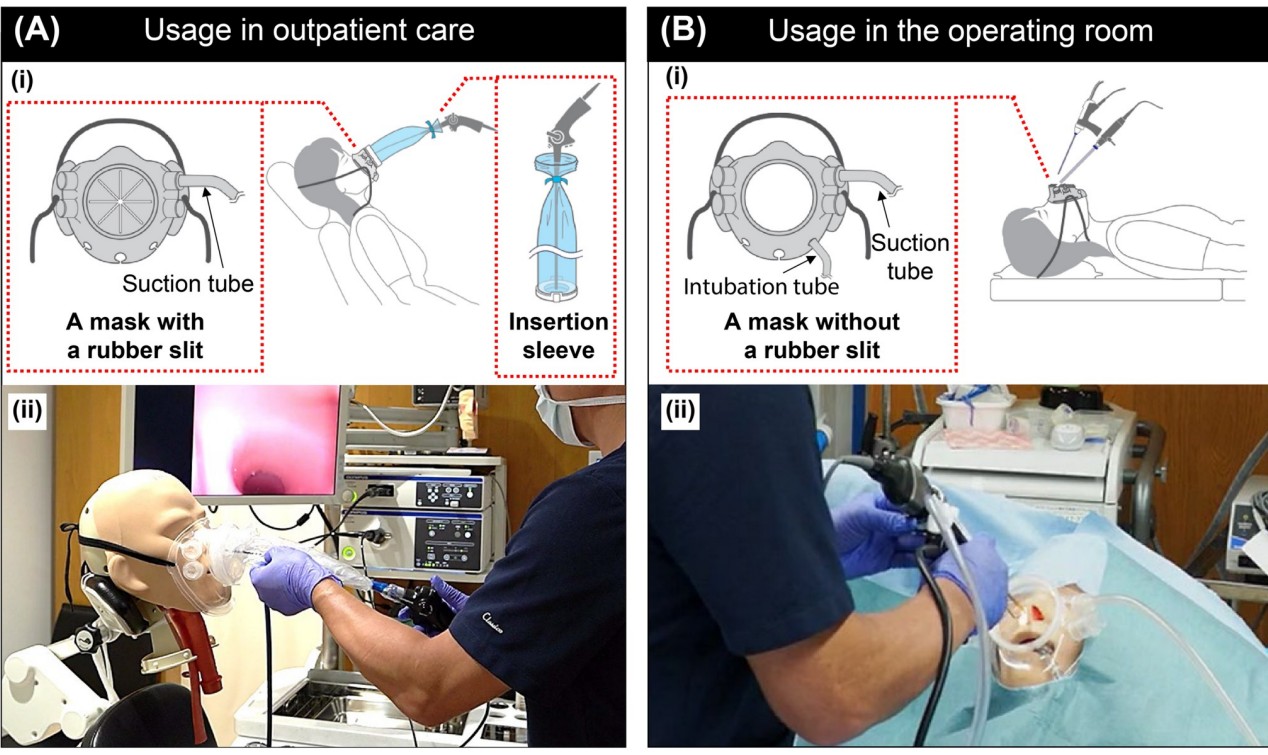

**Fig 2. Device usage in outpatient and surgical settings.** (A) Usage in outpatient care: a mask (for outpatient use) is attached to the head model, an insertion sleeve is attached to the endoscope, the joint of the insertion sleeve is joined to the mask, and the surgeon is examining the patient. (i) Illustration of the procedure. (ii) Image showing how the device is used. (B) Usage in the operating room (for surgery): a mask connected to a suction unit is attached to the head model, and the surgeon is performing the treatment. (i) Illustration of the procedure. (ii) Image showing how the device is used. The individual in this figure has given written informed consent (as outlined in PLOS consent form) to publish these case details.

mask. The endoscopist then manipulates the endoscope through the insertion sleeve (Fig 2A). When the endoscopic operation is completed, the endoscope is stored inside the insertion sleeve, and the insertion sleeve is removed from the mask.

When using a treatment instrument with a treatment endoscope, the instrument sleeve is attached to the instrument to be used beforehand and affixed to the endoscope. The control section cover is used to cover the control section of the endoscope. One end of the instrument sleeve is fixed to the endoscope, the instrument is inserted into the instrument channel, and the procedure is then performed. When the procedure is completed, the instrument is stored inside the instrument sleeve, which is then removed from the endoscope.

During surgery, the patient wears a mask without a rubber slit, and a suction tube is attached to the suction port of the mask. The surgeon inserts the operative tools through the insertion port in front of the mask and performs the procedure as usual (Fig 2B).

## Experiments

### Evaluation of the reduction of droplets and aerosols dispersed from patient model.
We created models that simulated droplets and aerosols dispersed from patients and evaluated the reduction of droplets and aerosols among the different device configurations. The device configurations common between the droplet and aerosol simulation models are as follows: Control (no device attached); No. 1: wearing a mask without a rubber slit (for surgery) without suction; No. 2: wearing a mask without a rubber slit (for surgery) with suction; No. 3: wearing

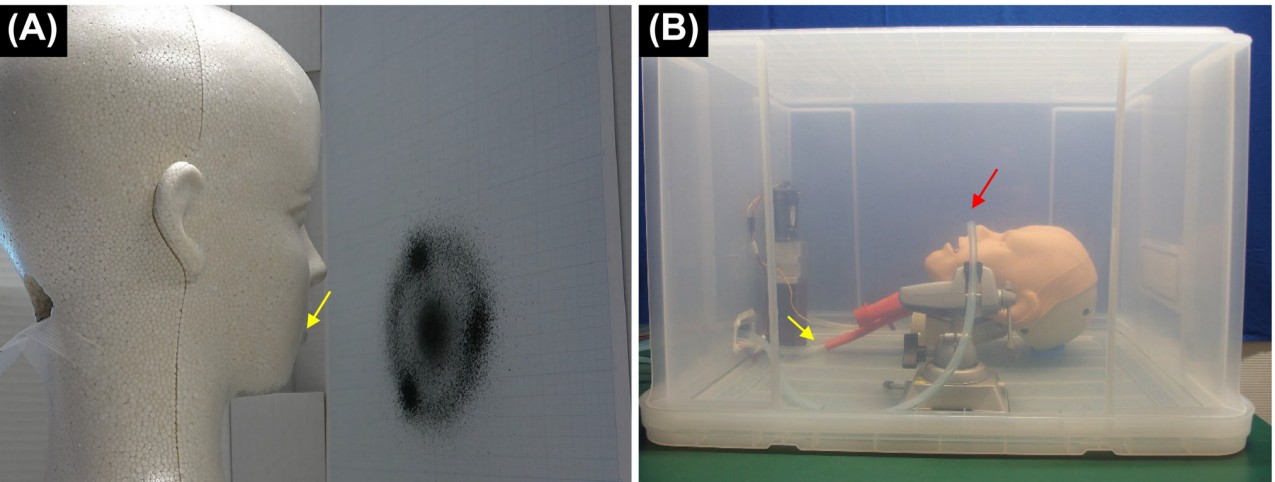

**Fig 3. Simulated droplet and aerosol replication models.** (A) A sprayer was installed in the mouth of the head model (yellow arrow) and ink was sprayed. The paper onto which the ink was sprayed was imaged, and the number of pixels that became black after binarization was measured. (B) The airway management training model was placed inside the plastic case and the smoke from an e-cigarette was sprayed through a tube from outside the plastic case (yellow arrow). The number of aerosol particles diffused into the plastic case was measured with a particle counter (red arrow).

a mask with a rubber slit (for outpatients) without suction; No. 4: wearing a mask with a rubber slit (for outpatients) with suction; No. 5: wearing a mask with a rubber slit (for outpatients) and using an endoscope (OLYMPUS ENF-VH2; Olympus Medical Systems Corporation, Tokyo, Japan) with an insertion sleeve without suction; and No. 6: wearing a mask with a rubber slit (for outpatients) and using an endoscope (OLYMPUS ENF-VH2; Olympus Medical Systems Corporation, Tokyo, Japan) with an insertion sleeve with suction.

**Droplet simulation model.** The droplets dispersed from patients' oral and nasal cavities were simulated with black ink. In a test simulating sneezing, it was reported that droplets were scattered to a distance of 66 cm [11]; thus, we created a model that could simulate the same flight distance. A sheet of paper was placed 10 cm in front of a sprayer installed in the mouth of a Styrofoam model head, and 0.2 mL of black ink was sprayed. The inked paper was scanned and imaged. The brightness value of each pixel was binarized at a threshold value at which the background and the attached ink could be separated, and each pixel was divided into black and white. The number of black pixels was measured five times under the same conditions, and the average number of black pixels was compared with different device conditions. For the devices with suction, suction was applied from the suction port at a pressure of -40 kPa and a flow rate of 40 L/min (Fig 3A).

**Aerosol simulation model.** Aerosols dispersed from patients' oral and nasal cavities were simulated by e-cigarette (FLEVO; GIEX, Tokyo, Japan) smoke. The e-cigarette was sprayed from the pulmonary side of an airway management training model (AirSim Advance Bronchi X; TruCorp., Lurgan, UK) so that cigarette smoke was only disseminated from the oral and nasal cavities. The e-cigarette was connected to a pump controlled by a computer to control the duration of the e-cigarette spray. The e-cigarette released smoke for 0.5 seconds in order to generate a similar number of aerosols as discharged in one cough [12, 17], and the number of e-cigarette smoke particles were similar to that discharged with a cough. The airway management training model was placed inside a plastic case in order to provide a stable environment.

A particle counter (MET ONE HHPC 6+; Beckman Coulter, Brea, CA) was used to measure aerosols emitted. The particle counter was set to measure the number of particles 0.3–1.0 μm in size for 120 seconds with a sampling interval of 1 second. The total number of

particles was defined as the total number of particles for 120 seconds at the time of evaluation minus the total number of particles for 120 seconds under normal conditions before evaluation. This calculation was performed in order to eliminate the influence of the number of particles present under normal conditions. The number of particles was measured five times under identical conditions, and the results were compared under varying conditions of wearing protective equipment and suction conditions as in the droplet experiment. For conditions with suction, suction was applied from the suction port at a pressure of -40 kPa and a flow rate of 40 L/min (Fig 3B).

**Evaluation of the reduction of droplets dispersed through the endoscope.**   We created two simulation models as given below.

*Simulation model of the body fluids dispersed during removal of a treatment device.* First, an instrument sleeve was placed on the procedure instrument. The instrument was inserted through the instrument channel, and the tip of the instrument was extended by several centimeters. The instrument sleeve was fixed to the endoscope. The tip of the endoscope was dipped into black ink and the instrument was then removed, following which the sleeve was checked for the presence of black ink adhering to its inner surface.

*Simulation model of body fluids discharged from the suction button through the endoscope.* First, the control section cover was attached to the endoscope. The endoscope was connected to an aspirator, which was set to a pressure of −66 kPa and a flow rate of 40 L/min. The tip of the endoscope was dipped into black ink and the suction button was pressed through the control section cover once every second for a total of 30 times. Thereafter, the adhesion of black ink inside the operator's cover was checked.

As no human or animal subjects were used in this study, ethical review board approval and study participation consent were not applicable. The individual in this manuscript has given written informed consent (as outlined in PLOS consent form) to publish these case details.

## Statistical analysis

Each condition was compared to the control and conditions with suction were compared to those without suction with respect to the average number of black pixels and aerosols, respectively. Statistical analyses were conducted using Minitab statistical software version 20 (Minitab LLC, State College, PA, USA) to assess differences in these comparisons. Nonparametric statistical techniques (Mann-Whitney U tests) were applied owing to the small sample size.

## Results

### Evaluation of the reduction of droplets and aerosols dispersed from patient model

**Droplet simulation model.**   No significant difference was observed in the number of black pixels (representing patient droplets) for device configuration No. 1 and No.2 (wearing a mask without a rubber slit) and those of the control. However, compared to the control, decreases of more than 95% and equal to 100% were observed for device configuration No. 3 and No.4 (wearing a mask with a rubber slit) and device configuration No. 5 and No.6 (wearing a mask with a rubber slit and using an endoscope with an insertion sleeve), respectively. No significant difference in the data between using devices with and without suction was observed (Fig 4A) (Table 1).

**Aerosol simulation model.**   There was a reduction in the number of aerosols in the absence of suction relative to the control; aerosols were reduced by 28% in No. 1 (wearing a mask without a rubber slit), by 42% in No. 3 (wearing a mask with a rubber slit), and by 63%

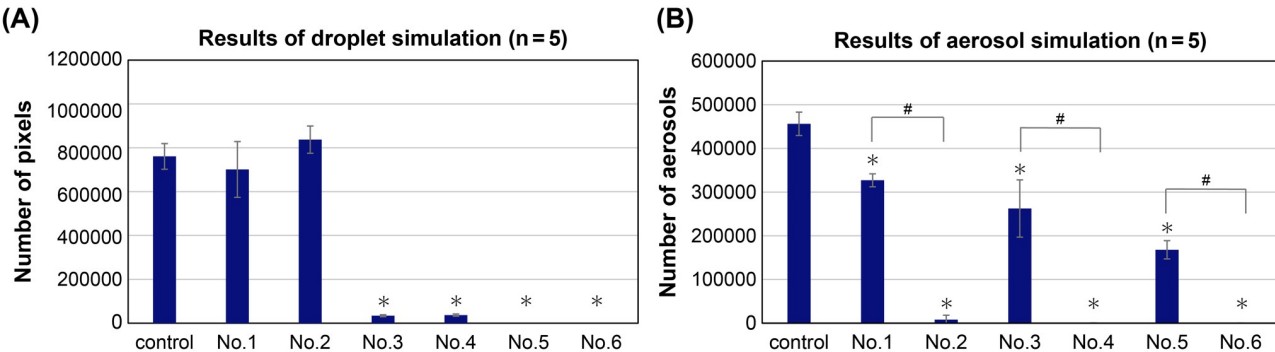

**Fig 4. Results of the droplet and aerosol simulations.** Experimental evaluation was performed under the following device configurations: control—no mask; No. 1—wearing a mask without a rubber slit (for surgery) without suction; No. 2—wearing a mask without a rubber slit (for surgery) with suction; No. 3—wearing a mask with a rubber slit (for outpatients) without suction; No. 4—wearing a mask with a rubber slit (for outpatients) with suction; No. 5—wearing a mask with a rubber slit (for outpatients) and using an insertion sleeve for the endoscope without suction; No. 6—wearing a mask with a rubber slit (for outpatients) and using an insertion sleeve for the endoscope with suction. (A) The average number of black pixels, representing droplets ($n = 5$). (B) The average number of aerosols ($n = 5$). * $P < 0.05$ vs. control. # $P < 0.05$ between devices used with suction and those used without suction. Statistical analysis was performed using the Mann-Whitney $U$ test.

or more in No. 5 (wearing a mask with a rubber slit and using an endoscope with an insertion sleeve). When suction was applied, there was a significant decrease of 98% or more in No. 2 (wearing a mask without a rubber slit), and 99% or more in No. 4 (wearing a mask with a rubber slit) and No. 6 (wearing a mask with a rubber slit and using an endoscope with an insertion sleeve) compared to that of the control. In addition, there was a significant difference in the data between the conditions with and without suction, indicating that the reduction was more pronounced when suction was present (Fig 4B) (Table 1).

### Evaluation of reduction of droplets dispersed through an endoscope

**Simulation model of the body fluids dispersed during removal of a treatment device.** The ink was confirmed to adhere to the inner surface of the instrument sleeve (Fig 5A).

**Simulation model of body fluid discharged from the suction button through the endoscope.** The ink was confirmed to adhere to the inner surface of the control section cover (Fig 5B).

## Discussion

In otorhinolaryngology practice, several methods of shielding patients or surgeons during endoscopic examination and treatment have been reported [1, 18–24]. For example, Farneti et al. created the shielded ear, nose, and throat (ENT) headlights and protective shields that could be fitted to examination chairs for outpatient visits in an ENT setting [20]. Viera-Artiles et al., using a 3D printer, developed a mask that could be attached to the nose to eliminate aerosols during endoscopic nasal surgery [22].

The device we developed in this study is intended for disposable use. Protective equipment should be disposable because reusable products require cleaning and disinfection. This device is premised on the use of PPE for normal medical examination and surgery.

Based on the results of the desk study, we will discuss the use and effects of the device in various situations in otorhinolaryngology.

First, we will consider its use in the operating room, where aerosols are expected to be generated during many procedures. During surgery, there are many situations in which

**Table 1. Results of the droplet and aerosol simulations.**

| No. | Mask | Rubber slit | Sleeve | Suction | Droplet (*n* = 5) | | | | | | | | | |
|---|---|---|---|---|---|---|---|---|---|---|---|---|---|---|
| | | | | | Average number of pixels (95% CI) | SD | Vs. control | | | | With suction vs. without suction | | | |
| | | | | | | | Average difference (95% CI) | % Decrease | *P* value | Effect size | Average difference (95% CI) | % Decrease | *P* value | Effect size |
| Control | - | - | - | - | 760761 (688089,833433) | 58528 | — | — | — | — | — | — | — | — |
| 1 | + | - | - | - | 701091 (542863,859319) | 127432 | -59670 (-220878, 101538) | -7.84 | 0.40 | 0.60 | 136101 (-26984, 299186) | 19.41 | 0.14 | -1.35 |
| 2 | + | - | - | + | 837192 (759790,914595) | 62337 | 76431 (-13991, 166854) | 10.04 | 0.09 | -1.26 | | | | |
| 3 | + | + | - | - | 33965 (28540, 39389) | 4369 | -726796 (-799670, -653923) | -95.53 | <0.05 | 17.51 | 2778 (-4221, 9776) | 8.17 | 0.83 | -0.59 |
| 4 | + | + | - | + | 36742 (30570,42915) | 4971 | -724019 (-796952, -651085) | -95.17 | <0.05 | 17.43 | | | | |
| 5 | + | + | + | - | 0 | N.A. | -760761 | -100 | <0.05 | 18.38 | 0 | 0 | N.A. | N.A. |
| 6 | + | + | + | + | 0 | N.A. | -760761 | -100 | <0.05 | 18.38 | | | | |
| No. | Mask | Rubber slit | Sleeve | Suction | Aerosol (*n* = 5) | | | | | | | | | |
| | | | | | Average number of aerosols (95% CI) | SD | Vs. control | | | | With suction vs. without suction | | | |
| | | | | | | | Average difference (95% CI) | % Decrease | *P* value | Effect size | Average difference (95% CI) | % Decrease | *P* value | Effect size |
| Control | - | - | - | - | 456267 (422998, 489535) | 26793 | — | — | — | — | — | — | — | — |
| 1 | + | - | - | - | 327019 (308714, 345324) | 14742 | -129248 (-162713, -95782) | -28.32 | <0.05 | 5.97 | -318982 (-337906, -300058) | -97.54 | <0.05 | 25.2 |
| 2 | + | - | - | + | 8037 (-4557, 20631) | 10143 | -448230 (-481165, -415295) | -98.23 | <0.05 | 22.12 | | | | |
| 3 | + | + | - | - | 262385 (180921, 343849) | 65608 | -193882 (-275352, -112411) | -42.49 | <0.05 | 3.86 | -262183 (-343647, -180718) | -99.92 | <0.05 | 5.65 |
| 4 | + | + | - | + | 202 (-142, 547) | 278 | -456064 (-489335, -422794) | -99.95 | <0.05 | 24.07 | | | | |
| 5 | + | + | + | - | 167871 (141787, 193954) | 21007 | -288396 (-324400, -252392) | -63.20 | <0.05 | 11.97 | -167685 (-193769, -141601) | -99.89 | <0.05 | 11.28 |
| 6 | + | + | + | + | 185 (-48, 418) | 188 | -456081 s (-489351, -422812) | -99.95 | <0.05 | 24.07 | | | | |

Under each condition, the average number of black pixels/aerosols, the percent reduction compared with that of the control group, the *P* value and effect size compared with those of the control group (Mann-Whitney *U* test), and the comparison of *P* values and effect sizes between devices used with and without suction (Mann-Whitney *U* test) are indicated.

endoscopes and debriders are inserted into the patient's nasal cavity simultaneously, and operability is the most important factor. The mask we developed has an opening for the insertion of an endoscope, which allows endoscopes to be operated in the same way as in normal medical treatment. In addition, by connecting the mask to an aspirator, we were able to reduce

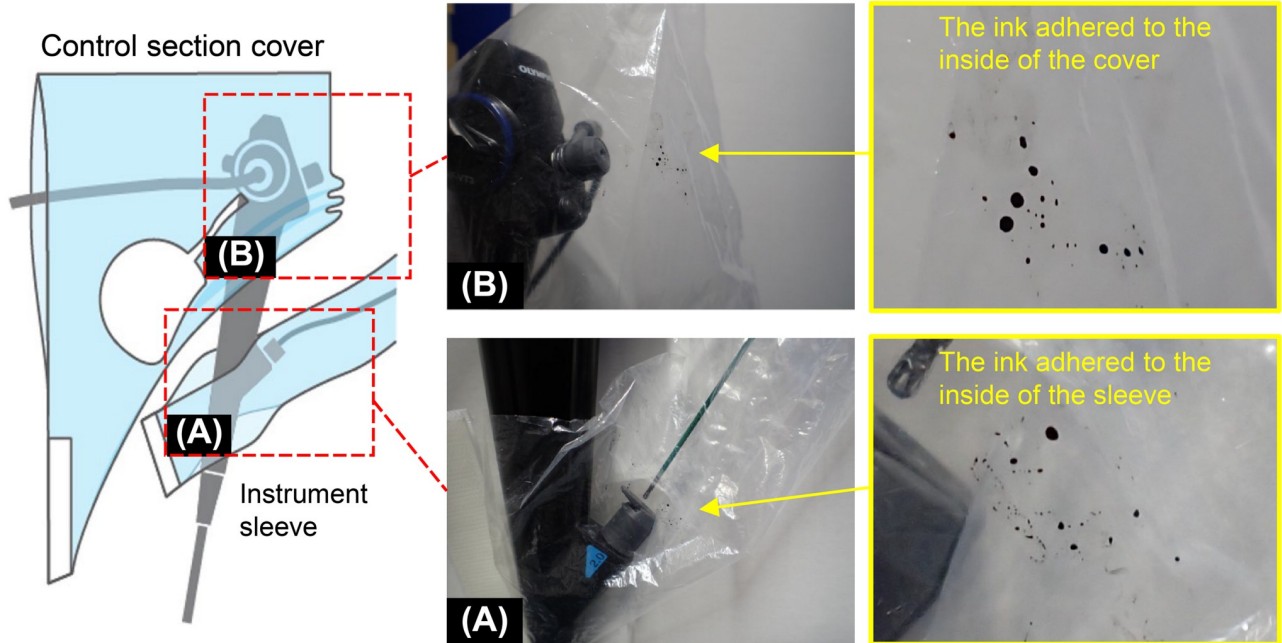

**Fig 5. Evaluation of instrument sleeve and control section cover.** (A) Ink that is released when the instrument is removed from the instrument channel of the endoscope adheres to the inside of the instrument sleeve (yellow arrow). An enlarged view of the adhered area is shown (yellow box). B: Ink released when pressing the suction button on the endoscope control unit adheres to the inside of the control section cover (yellow arrow). An enlarged view of the adhered area is shown (yellow box).

aerosols by as much as 98%. However, it is difficult to protect against droplets because the endoscope insertion port is open, but it is unlikely that droplets caused by coughing will spread when patients are under general anesthesia during surgery.

In outpatient clinics where patients often cough, a mask with a rubber slit reduced droplets by more than 95%. In particular, connecting the mask to a suction device reduced aerosols by more than 99%. Although a 42% reduction in aerosols was achieved without suction, suction is considered preferable in view of its high effectiveness in reducing aerosols. When inserting an endoscope, diffusion of droplets and aerosols can be suppressed by more than 99% by fitting the endoscope with an insertion sleeve. Furthermore, the sleeve is expected to prevent contact with any body fluids of patients that adhere to the inserted part of the endoscope after the examination.

When an endoscope is used for treatment, it was shown that the attachment of the instrument sleeve to the instrument in advance prevents contact with and diffusion of body fluid dispersed by removal of the instrument. In addition, our results indicate that we can expect that leakage of droplets of fluid from the suction button can be reduced by attaching the control section cover.

There are, however, some limitations to this device and our experiments. In this experiment, we simulated aerosols using e-cigarette smoke and droplets with black ink; however, details regarding the generation of aerosols and droplets and the route of infection in actual clinical use are unknown. This device aims to reduce the risk of transmission of infectious diseases but does not guarantee infection prevention. We plan to verify the effectiveness of this device in clinical practice. In addition, since the mask is worn by the patient, we plan to assess the patient's normal breathing and level of discomfort when wearing the mask.

It is expected that this device may be used not only in otorhinolaryngology but also during endoscopic procedures in other fields, such as gastroenterology and respiratory medicine. As the importance of infection prevention measures for medical personnel has increased with the outbreak of COVID-19, we expect that this device may contribute to the prevention of infection in many endoscopic fields.

## Conclusions

In this study, we developed a device for infection control in collaboration with a gastrointestinal endoscopist and Olympus Medical Systems Corporation. The device was found to be effective in reducing the spread of droplets and aerosols. Although there are some limitations related to the theoretical nature of this study, we expect this device to be effective in reducing droplet and aerosol diffusion during outpatient and surgical procedures in otorhinolaryngology.

## Author Contributions

**Conceptualization:** Yoshinori Matsuwaki, Daiki Ariyoshi, Daisuke Kikuchi.

**Formal analysis:** Daiki Ariyoshi, Risa Iwamoto.

**Investigation:** Daiki Ariyoshi, Tetsuya Morishima, Keita Kimura.

**Methodology:** Daiki Ariyoshi, Tetsuya Morishima, Keita Kimura.

**Project administration:** Daiki Ariyoshi.

**Resources:** Yoshinori Matsuwaki, Daiki Ariyoshi, Risa Iwamoto, Tetsuya Morishima, Keita Kimura.

**Supervision:** Yoshinori Matsuwaki, Daiki Ariyoshi.

**Validation:** Daiki Ariyoshi, Tetsuya Morishima, Keita Kimura.

**Visualization:** Daiki Ariyoshi, Risa Iwamoto.

**Writing – original draft:** Daiki Ariyoshi, Risa Iwamoto.

**Writing – review & editing:** Yoshinori Matsuwaki, Daiki Ariyoshi.

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
