## [Decision Letter · Decision Letter 0]

28 Feb 2022

PONE-D-21-36009Development of protective equipment for endoscopic treatment and surgery in otorhinolaryngologyPLOS ONE

Dear Dr. Matsuwaki,

Thank you for submitting your manuscript to PLOS ONE. After careful consideration, we feel that it has merit but does not fully meet PLOS ONE’s publication criteria as it currently stands. Therefore, we invite you to submit a revised version of the manuscript that addresses the points raised during the review process.

Please pay attention to more clearer figures and results

We look forward to receiving your revised manuscript.

Kind regards,

Claudio Andaloro

Academic Editor

PLOS ONE

Journal Requirements:

I have read the journal's policy and the authors of this manuscript have the following competing interests: 

This research and development endeavor was conducted in collaboration with Olympus Medical Systems Corporation and a gastrointestinal endoscopist (Dr. Daisuke Kikuchi). Matsuwaki Clinic Shinagawa and Olympus Medical Systems Corporation have concluded an advisory agreement regarding this development. 

3. We note that Figure 2 includes an image of a participant in the study. 

Reviewers' comments:

Reviewer's Responses to Questions

**Comments to the Author**

1. Is the manuscript technically sound, and do the data support the conclusions?

Reviewer #1: Yes

Reviewer #2: Yes

2. Has the statistical analysis been performed appropriately and rigorously? 

Reviewer #1: Yes

Reviewer #2: Yes

3. Have the authors made all data underlying the findings in their manuscript fully available?

Reviewer #1: Yes

Reviewer #2: No

4. Is the manuscript presented in an intelligible fashion and written in standard English?

Reviewer #1: Yes

Reviewer #2: Yes

5. Review Comments to the Author

Reviewer #1: This study is indeed essential and relevant to be carried out. The paper was well written with acceptable figures. My only concern is that the actual p values were not stated, either in the text or figure legends. Additionally, since the sample size was small, including an effect size analysis is beneficial to support the p values.

Reviewer #2: This manuscript is the study to validate the protective equipment for endoscopic 'treatment and surgery'. The simulated situations are droplet spread and aerosal simulation model. The protective equipment are: suction-capable mask, sleeves and cover for suction. The main result is reduction of droplet and aerosal diffusion from there protective equipment.

There are some issues needed to be elucidated.

1) The main result has been displayed in two bar graphs but not in the table. The actual parameter/value will be easier and clearer for verification of 95% Confidence interval and p-value.

2) The author mention about the number of test as 'five times' but these values have not been shown in the figures.

3) The title mention about 'endoscopic treatment and surgery' but the testing situation was conducted by 'flexible endoscope' which unusual for 'surgery' in ENT.

4) The displayed pictures are difficult for reader to follow the method. More simplified, eg cartoon/drawing, should be done for clarification.

6. PLOS authors have the option to publish the peer review history of their article (what does this mean?). If published, this will include your full peer review and any attached files.

Reviewer #1: No

Reviewer #2: No

---

## [Author Response · Author response to Decision Letter 0]

19 Apr 2022

Response to Reviewers’ Comments

Reviewer #1: This study is indeed essential and relevant to be carried out. The paper was well written with acceptable figures. My only concern is that the actual p values were not stated, either in the text or figure legends. Additionally, since the sample size was small, including an effect size analysis is beneficial to support the p values.

Reply: 

Thank you for your encouraging comment.

I have added p-values to the legend of Fig.4 and p-values and the effect sizes in Table 1 in the revised manuscript (lines 243-256).

Reviewer #2: This manuscript is the study to validate the protective equipment for endoscopic 'treatment and surgery'. The simulated situations are droplet spread and aerosol simulation model. The protective equipments are: suction-capable mask, sleeves and cover for suction. The main result is reduction of droplet and aerosol diffusion from there protective equipment.

There are some issues needed to be elucidated.

1) The main result has been displayed in two bar graphs but not in the table. The actual parameter/value will be easier and clearer for verification of 95% Confidence interval and p-value.

Reply: 

Thank you for your comment.

I have added p-values to the legend of Fig.4. I have also added 95% confidence interval values and p-values as well as the effect sizes to Table 1 in the revised manuscript (lines 243-256)

2) The author mention about the number of test as 'five times' but these values have not been shown in the figures.

Reply: 

Thank you for pointing this out.

I have indicated the number of tests as “n=5” in the revised version of Fig. 4 and in Table 1 (lines 243-245). I have also added it to the legend of Fig. 4 (lines 246-256).

3) The title mention about 'endoscopic treatment and surgery' but the testing situation was conducted by 'flexible endoscope' which unusual for 'surgery' in ENT.

Reply: 

Thank you for your constructive feedback.

During surgery, a rigid scope is used. Since the patient is anesthetized during surgery, the risk of splashing due to the patient sneezing, etc., is low. Therefore, the insertion sleeve need not be attached to the rigid scope or treatment tool, and only the surgical mask should be used.

During outpatient treatments, in addition to the outpatient mask, the insertion sleeve is attached to the flexible endoscope or the rigid endoscope and used in combination to reduce the risk of splashing.

In the experiment, the surgical scene (Nos. 1 and 2 in Fig. 4) and the outpatient scene (Nos. 3-6 in Fig. 4) are shown. The flexible endoscope is used only for evaluation in outpatient situations. This content is described in the Experiments section on (lines 145-156).

4) The displayed pictures are difficult for reader to follow the method. More simplified, eg cartoon/drawing, should be done for clarification.

Reply: 

Thank you for this valuable suggestion.

Cartoon drawings have been added to Fig. 2 to help the readers understand the outpatient and surgical procedures. I hope that this will also help address the query raised in Comment 3. I also added a cartoon drawing to Fig. 5 to clearly present the experimental procedures.

---

## [Decision Letter · Decision Letter 1]

12 May 2022

Development of protective equipment for endoscopic treatment and surgery in otorhinolaryngology

PONE-D-21-36009R1

Dear Dr. Matsuwaki,

We’re pleased to inform you that your manuscript has been judged scientifically suitable for publication and will be formally accepted for publication once it meets all outstanding technical requirements.

Kind regards,

Claudio Andaloro

Academic Editor

PLOS ONE

Additional Editor Comments (optional):

Reviewers' comments:

Reviewer's Responses to Questions

**Comments to the Author**

1. If the authors have adequately addressed your comments raised in a previous round of review and you feel that this manuscript is now acceptable for publication, you may indicate that here to bypass the “Comments to the Author” section, enter your conflict of interest statement in the “Confidential to Editor” section, and submit your "Accept" recommendation.

Reviewer #1: All comments have been addressed

Reviewer #2: All comments have been addressed

2. Is the manuscript technically sound, and do the data support the conclusions?

Reviewer #1: Yes

Reviewer #2: Yes

3. Has the statistical analysis been performed appropriately and rigorously? 

Reviewer #1: Yes

Reviewer #2: Yes

4. Have the authors made all data underlying the findings in their manuscript fully available?

Reviewer #1: Yes

Reviewer #2: Yes

5. Is the manuscript presented in an intelligible fashion and written in standard English?

Reviewer #1: Yes

Reviewer #2: Yes

6. Review Comments to the Author

Reviewer #1: All my concerns have been satisfactorily addressed by the authors and the paper looks much better now. As such, this paper can be accepted for publication.

Reviewer #2: The manuscript has been revised according to the reviewer comments. The comments have been reposed clearly point-by-point.

7. PLOS authors have the option to publish the peer review history of their article (what does this mean?). If published, this will include your full peer review and any attached files.

Reviewer #1: No

Reviewer #2: **Yes: **Pongsakorn Tantilipikorn MD PhD

---

## [Editor Report · Acceptance letter]

3 Jun 2022

PONE-D-21-36009R1 

Development of protective equipment for endoscopic treatment and surgery in otorhinolaryngology 

Dear Dr. Matsuwaki:

I'm pleased to inform you that your manuscript has been deemed suitable for publication in PLOS ONE. Congratulations! Your manuscript is now with our production department. 

Kind regards, 

on behalf of

Dr. Claudio Andaloro 

Academic Editor

PLOS ONE